# Single Nucleotide Polymorphisms in Genes Encoding Toll-Like Receptors 7 and 8 and Their Association with Proviral Load of SRLVs in Goats of Polish Carpathian Breed

**DOI:** 10.3390/ani11071908

**Published:** 2021-06-26

**Authors:** Monika Olech, Katarzyna Ropka-Molik, Tomasz Szmatoła, Katarzyna Piórkowska, Jacek Kuźmak

**Affiliations:** 1Department of Biochemistry, National Veterinary Research Institute, 24-100 Pulawy, Poland; jkuzmak@piwet.pulawy.pl; 2Department of Animal Molecular Biology, National Research Institute of Animal Production, Krakowska 1, 32-083 Balice, Poland; katarzyna.ropka@izoo.krakow.pl (K.R.-M.); tomasz.szmatola@izoo.krakow.pl (T.S.); katarzyna.piorkowska@izoo.krakow.pl (K.P.); 3Center for Experimental and Innovative Medicine, University of Agriculture in Krakow, Rędzina 1c, 30-248 Krakow, Poland

**Keywords:** small ruminant lentiviruses (SRLVs), toll-like receptor 7 (TLR7), toll-like receptor 8 (TLR8), proviral load, single nucleotide polymorphisms (SNPs)

## Abstract

**Simple Summary:**

The purpose of the study was to identify SNPs in genes encoding TLR7 and TLR8 in goats of Carpathian breed and analyze their association with the SRLVs provirus concentration. A total of 14 SNPs were detected, 6 SNPs in the TLR7 gene locus and 8 SNPs in the TLR8 gene. These SNPs were located in intron, 3′UTR and 5′UTR regions and within the coding sequences leading to the synonymous mutations. Our results revealed that 9 out 14 identified polymorphisms were associated with the SRLVs proviral concentration. This finding supports a role for genetic variations of TLR7 and TLR8 in SRLVs infection.

**Abstract:**

Toll-like receptors (TLRs) 7 and 8 are important in single-stranded viral RNA recognition, so genetic variation of these genes may play a role in SRLVs infection and disease progression. Present study aimed to identify SNPs in genes encoding TLR7 and TLR8 in goats of Carpathian breed and analyze their association with the SRLVs provirus concentration as index of disease progression. A total of 14 SNPs were detected, 6 SNPs in the TLR7 gene locus and 8 SNPs in the TLR8 gene. Nine of the 14 identified polymorphisms, 4 in the TLR7 gene and 5 in TLR8 gene, were significantly associated with the SRLVs proviral concentration. These SNPs were located in 3′UTR, 5′UTR and intron sequences as well as in the coding sequences, but they led to silent changes. Homozygous genotypes of three TLR7 SNPs (synonymous variant 1:50703293, 3′UTR variant 1:50701297 and 5′UTR variant 1:50718645) were observed in goats with lower provirus copy number as well as in seronegative animals. The results obtained in this study suggest that SNPs of TLR7/TLR8 genes may induce differential innate immune response towards SRLVs affecting proviral concentration and thereby disease pathogenesis and progression. These findings support a role for genetic variations of TLR7 and TLR8 in SRLVs infection and warrants further studies on the effect of TLR7/TLR8 polymorphisms on SRLVs infection in different populations.

## 1. Introduction

Maedi visna virus (MVV) and caprine arthritis encephalitis virus (CAEV) also referred as small ruminant lentiviruses (SRLVs) are two related retroviruses which infect sheep and goats. These viruses infect monocytes, macrophages and dendritic cells and despite immune response cause a lifelong infection which can persist for months in latent or subclinical form. The most prevalent clinical signs as an outcome of SRLVs are associated with arthritis, neurological disorders, mastitis, emaciation and pneumonia [1]. Transmission occurs from infected dams to offspring by colostrum/milk consumption and between adults mainly through respiratory secretion [2]. Since no effective vaccines are available, the current control strategies against SRLVs infections are mostly based on the detection and culling of infected animals [3]. What is more, SRLVs infections have become a worldwide problem bringing considerable financial losses in the small ruminant industry [3,4,5]. In Poland, any SRLVs control programs have been never implemented, and thus, infections with SRLVs are quite common. The overall true prevalence at the flock level reached 33.3% and 71.9% in sheep and goats, respectively [6,7].

In the course of SRLVs infection, their genome is integrated into the host genome in the form of proviral DNA which load is a factor determining disease prediction. It was shown that high proviral load corresponded to higher lesion score [8,9]. On the other hand, the presence of animals with low SRLVs proviral load may suggest the implication of host factors that may restrict and control viral replication. Such animals which are also referred as long term non-progressor show competent humoral immune response in the absence of virus replication leading to lower SRLVs transmission [10,11,12]. This different reactivity to SRLVs infection may suggest different host immune response ability to control this infection. 

In SRLVs-infected animals, both innate and adaptive immunity are induced. Moreover, several genes associated with resistance/susceptibility to SRLVs infection and disease outcome have been identified [13,14]. Toll-like receptors (TLRs) are host pattern recognition receptors (PRRs) that play a pivotal role in the innate immune system. PRRs activate the innate immune system through different signaling pathways upon recognition of variety of structurally conserved molecules derived from pathogens known as PAMPs (pathogen-associated molecular patterns). TLRs are type I transmembrane proteins which contain three domains, an ectodomain (ECD) containing leucine-rich repeat (LRR) motifs that mediate PAMPs recognition and a cytoplasmic Toll/interleukin-1 receptor (TIR) domain linked by a single transmembrane (TM) domain [15]. TLR involved in response to viral infection are TLR1-TLR4 and TLR6-TLR9 [16]. In retroviral infections, TLR7 and TLR8, which share a high degree of structural similarity and recognize viral ssRNA (single-stranded RNA) within endosomal compartments, are of particular interest [17]. Activation of these TLRs initiates the MyD88-dependent pathway, mainly in NF-κB activation for inducing the expression of pro-inflammatory cytokines, chemokines, and type I and type III interferons [18,19]. It is well known that cytokines impairing expression is a key point in SRLVs-dependent immunity, pathogenies, and appearance of lesions [20,21]. However, so far, the role of lentivirus-induced TLR signaling has not been widely studied in small ruminants. It was showed that in the course of SRLVs infection, TLR7 and TLR8 become activated, inducing IFN-α, IL-6, TNF-α production and expression of antiviral proteins [21]. Upregulation of TLR7 and TLR8 was noted in naturally SRLVs-infected sheep showing lung lesions [13]. 

Single nucleotide polymorphisms (SNPs) of genes encoding proteins involved in innate response have aroused much attention, and a number of studies have been conducted to identify SNPs in these genes in different species [22,23,24]. It has been found that genetic variants of TLRs can be associated with different outcome of several diseases [15,25]. Polymorphisms within TLR7 and TLR8 have been linked to susceptibility and progression of different viral diseases including these caused by Human Immunodeficiency virus (HIV-1), Hepatitis C virus (HCV), Chikungunya virus (CHIKV) and Crimean-Congo hemorrhagic fever (CCHF) virus [26,27,28,29]. Mikula et al. [30,31] suggested that mutations in TLR7, TLR8 and TLR9 may play an important role as host factor predisposing sheep for infection with SRLVs. Despite these observations, there is currently a lack of any studies reporting polymorphisms of caprine genes encoding TLR7 and TLR8 and their possible role in SRLVs infection. Herein, this study was conducted to identify SNPs in genes encoding TLR7 and TLR8 in goats of Carpathian breed and analyze their association with the SRLVs infection and provirus concentration. We focused on goats of Carpathian breed because represent a remnant population of an ancient breed; thus, analysis in this population could be important for its conservation.

## 2. Materials and Methods

### 2.1. Animals and Sample Preparation

The study was performed in one flock, counting 32 adult goats representing Carpathian breed. All 32 goats were examined in this study. The goats of Carpathian breed were widely found in Carpathian Mountain in Poland in the 19th and 20th century and then became an extinct breed. In 2005, all of these goats were moved to the National Research Institute of Animal Production in Cracow where this flock has been recreated and currently is covered by the genetic resources protection program, supported by the Ministry of Agriculture and Rural Development; however, its risk of extinction is high. 

All goats were clinically healthy and maintained at the same environmental and feeding conditions. Age of goats ranged from 2 to 10 years (average ~6 years). Blood samples were taken by jugular venipuncture in EDTA tubes and in second tubes for serum collection, and the blood was collected from all animals at the same day. DNA extraction was performed using Sherlock AX isolation kit (A&A Biotechnology, Gdynia, Poland) and quality of preparation was checked using NanoDrop 2000 (Thermo Fisher Scientific, Waltham, MA, USA). Status of goats for SRLVs infection was confirmed by serological testing using ELISA (ID Screen MVV/CAEV Indirect Screening test, IDVet, France), according to the manufacturer’s recommendations. All procedures associated with animal handling and treatments were approved by the Local Ethical Committee on Animal Testing at the University of Life Sciences in Lublin (Poland).

### 2.2. SNP Identification–Variant Calling

As a first approach, the screening for SNPs identification was performed using already available RNA sequencing (RNA-seq) data obtained from 12 goats from tested flock (data available under GEO GSE168160 accession number). For SNPs identification, the quality of raw data was measured with the use of FastQC software [32] which was followed by trimming procedure that focused on removing adapter content, reads of low quality (phred quality < 20) and low read length (minimal read length set to 35) with the use of Trimmomatic software [33]. Then, the quality of trimmed reads was checked again to ensure the effectiveness of trimming. The next step was mapping procedure which was maintained with the use of Tophat software [34] to Capra Hircus ARS1 genome. Then, the duplicates were marked with the use of MarkDuplicates function of Picard Tools, and finally, the variant calling procedure was utilized with the use of Freebayes software [35] with a minimum coverage threshold set to 5. The filtration of the obtained variants was done with the use of VCFtools software [36] with the following parameters: minimum coverage set to 10; minimum quality set to 30; minimum combined coverage set to 120 and minimum combined quality set to 360. The annotation of the TLR8 and TLR7 gene variants was done with the use of Ensembl Variant Effect Predictor online software [37].

### 2.3. SNP Genotyping

Based on information on RNA-seq SNPs, for both TLR8 and TLR7 loci, the primers span gene’s regions containing polymorphisms were designed and PCR amplification, Sanger sequencing and SNPs analysis were performed on samples of all 32 goats. The primers span gene’s region containing SNPs was designed using Primer3 (v. 0.4.0) based on the reference sequence shown in Table 1 and Appendix A. The PCR amplification products were obtained using AmpliTaq Gold™ 360 Master Mix (Thermo Fisher Scientific, Waltham, MA, USA) and purified using enzyme mixture-EPPiC (A&A Biotechnology, Gdynia, Poland). The Sanger sequencing was performed using BigDye™ Terminator v3.1 Cycle Sequencing Kit (Thermo Fisher Scientific) and BigDye XTerminator™ Purification Kit (Thermo Fisher Scientific) according to the manufacturer’s protocol. The amplicons were capillary sequenced on 3500xL Genetic Analyzer (Applied Biosystems, Thermo Fisher Scientific, Waltham, MA, USA). Data were analyzed using Data Collection Software (Applied Biosystems). All identified variants were submitted to European Variation Archive (EVA) and received the Project ID PRJEB42246.

### 2.4. Poviral Load Quantification

DNA extracted from peripheral blood leukocytes (PBLs) [38] was quantified by the real time PCRs using Rotor-Gene Q Series ver. 2.0.3 (Qiagen) with primers and probe specifically designed for SRLV A5 subtype, which circulation in this flock was confirmed [39]. Sequence of forward and reverse primers and probe were CA5F (5′ TGGGAGTAGGACAAACAAATCA 3′), CA5R (5′ TGACATAT GCCTTACTGCTCTC 3′) and CA5P (5′ 6-FAM-TCACCCATTGTAGGCATAGCTGCC-BHQ-1 3′), respectively. A reference plasmid encompassing the target *gag* region was generated by the cloning of a 625 bp fragment into pDrive plasmid used to generate a standard curve based on 10-fold serial dilutions of plasmid DNA from 10^8^ to 1^0^. Amplification was performed in a total volume of 20 μL, according to the following cycling conditions: initial incubation and polymerase activation at 95 °C for 15 min and followed by 45 cycles of 94 °C for 60 s and 60 °C for 60 s. The reaction mixture for each PCR test contained 10 μL 2× QuantiTect Multiplex NoROX PCR buffer (Qiagen, Hilden, Germany), 400 nM of each of the primers, 200 nM of the specific probe, 5 μL of extracted genomic DNA. A non-template control (DEPC H_2_O) was included in each run. All samples were tested in duplicate, and the results were expressed as a mean copy number of provirus per 500 ng of genomic DNA of each goat. 

### 2.5. Statistical Analysis

To identify association between genotypes and proviral load, a simple linear model approach was performed using *t*-test. In this analysis SNP was used as a classification variable and proviral load values were used as analysis variables. Quantitative results given as means ± SD presented specific significant differences between animals within given genotype groups. The association between SRLVs DNA proviral load and age of tested goats were estimated using the Kruskal–Wallis H test. For this purpose, the animals were divided into 4 groups: 1 (2–3 years old), 2 (4–6 years old), 3 (7–8 years old) and 4 (9–10 years old). *p*-value of <0.05 was considered statistically significant. All statistical analyses were performed using SAS Enterprise (Statistical Analysis System, Version 8.02, 2001, Wadowice, Poland).

The genotype distribution of the SNPs of TLR7 and TLR8 were tested for deviation from the Hardy-Weinberg equilibrium (HWE) by means of χ^2^ analysis using Court lab- HW calculator. *p*-value of <0.05 were considered statistically significant. Phase program was used to assess frequency of haplotype distribution, and Haploview program [40] was used (default mode) to generate haplotype block structures and calculate LD values between SNPs.

## 3. Results 

### 3.1. Serology and qPCR

Out of 32 goats used in this study, 29 were found seropositive by ELISA and positive to quantitative polymerase chain reaction (qPCR) confirming the infection with SRLVs. Three goats were negative in both ELISA and qPCR. The average number of proviral copies in positive samples varied from 1 to 263 per 500 ng of genomic DNA. The correlation between proviral load and age of goats was not statistically significant (H = 6.893613; *p* < 0.0754; Kruskal-Wallis test).

### 3.2. TLR7 Gene-Detected SNPs, Allele and Haplotype Frequency

The variant calling method allowed to identify four polymorphisms in TLR7 gene locus. Two of them were 3′UTR (untranslated regions) variants (C/T 1:50701297 and T/C 1:50702074), one 5′UTR variant (C/T 1:50718645) and one synonymous polymorphism (T/C 1: 50703293) (Table 2). Additionally, Sanger sequencing performed on specimen collected from all goats revealed two additional SNPs, one in promoter and second in intron 1 regions (G/A 1: 50718760 and C/T 1: 50718466), which was not found in the first approach when whole blood transcriptomes were sequenced (Table 3).

The allele frequency analysis showed that two of all analyzed SNPs (promoter G/A 1:50718760 and 3′UTR T/C 1:50702074) were monomorphic but showed the mutant allele compared to the reference sequence. For the rest four polymorphisms, three genotypes were identified. The genotype and allele distributions of these four SNPs were shown in Table 4.

The Synonymous Variant (SV) T/C; 3′UTR and 5′UTR polymorphisms showed the similar allele distribution. These SNPs were most abundantly present in heterozygous goats (Table 4). Distribution of CC and TT genotype was from 16% to 31% and from 23% to 31%, respectively. For 3′UTR (C/T 1:50701297) and 5′UTR SNPs (C/T 1:50718645), only five animals, including 3 seronegative goats, carried CC genotype. For SV T/C 1:50703293 polymorphism, these goats carried TT genotype. The frequency of intron 1 (C/T 1:50718466) allele showed the predominance of CC genotype (91%), while opposite homozygotes TT and heterozygotes CT were 3% and 6%, respectively. One out 3 seronegative goats carried TT genotype while 2 other goats carried CC genotype. Moreover, the analyzed population was not in Hardy-Weinberg equilibrium according to intron C/T 1:50718466 (Table 4). Haplotype analysis allowed to detect 8 haplotype regions. Three haplotypes represented frequency from 10% to 51% while two haplotypes had frequency lower that 1% (Table 5). The linkage disequilibrium (LD) analysis showed the presence of one LD block and strong LD between four SNPs (Appendix A).

### 3.3. TLR8 Gene-Detected SNPs, Allele and Haplotype Frequency

In TLR8 gene, using RNA-seq, six polymorphisms were detected from which two were the 3′UTR variants and the rest were the synonymous SNPs (Table 2). Additionally, Sanger sequencing allowed detecting two extra SNPs in the 3′UTR region: (1:50659346 C/A, 1:50659136 A/C) (Table 3).

In TLR8 locus, 3′UTR T/C 1:50659202 SNP was monomorphic but showed the mutant allele compared to the reference sequence. For 1:50664682 A/G SNP, two genotypes were identified. The frequencies of identified AA and GT genotypes were 95% and 5%, respectively. For the rest six polymorphisms, three genotypes were identified. One type of homozygote (AA or CC) was found in most goats, including also seronegative animals while heterozygotes goats accounted for 32–42% (Table 6).

The haplotype analysis revealed the presence of 10 haplotypes from which frequency ranged from 29.7% to 13.30% (Table 5). The linkage disequilibrium (LD) analysis showed the one LD block involving 6 polymorphisms (Appendix A).

### 3.4. Association between SNPs and Provirus Copy Number

Because only three goats were uninfected with SRLVs, it was impossible to create control group with an equivalent number of animals to those of serologically positive and compare allele and genotype frequencies between infected and uninfected goats. Therefore, only associations between identified SNPs and provirus copy number of SRLVs were estimated for both TLR genes. 

It was showed that nine of the identified polymorphisms showed significant association with provirus copy number. For TLR7 gene, four polymorphisms were significantly associated with SRLVs provirus copy number (Figure 1). For TLR7 intron variant 1:50718466, goats with CC genotype had significantly higher proviral load that CT genotype goats (*p* value < 0.05). For three other SNPs (synonymous variant 1:50703293, 3′UTR variant 1:50701297 and 5′UTR variant 1:50718645), the heterozygote goats CT were characterized by higher provirus copy number (*p* < 0.05).

For TLR8 gene, five polymorphisms were significantly associated with provirus copy number (Figure 2). For TLR8 1:50666071 synonymous variant, homozygotes GG goats had lower proviral load than heterozygotes AG goats (*p* < 0.05). For two synonymous TLR8 polymorphisms (1:50664064 and 1:50664208), the goats with the CC genotype showed the lower copy number of SRLVs compared to CT animals. For 3′UTR variant 1:50659346 AC genotype, goats had significantly higher proviral load than CC genotype goats (*p* value < 0.05). The exact opposite was the case with 1:50659136 SNP, for which homozygotes AA goats had significantly higher SRLVs proviral load than both other genotypes (*p* < 0.01).

## 4. Discussion

The present study is the first attempt showing the genetic variability in TLR7 and TLR8 genes in SRLVs infected goats as well as association of SNPs with SRLVs proviral concentration.

We identified 6 polymorphisms in the TLR7 gene locus and 8 SNPs in the TLR8 gene. These SNPs were located in intron, 3′UTR and 5′UTR regions and within the coding sequences leading to the synonymous mutations which do not result in any amino acid change in encoded protein, due to genetic code redundancy. Synonymous substitutions and mutations, affecting noncoding DNA regions, are often considered silent mutations; however, it is not always the case that the mutation is silent. This kind of polymorphisms can produce different effects on gene expression leading to functional differences of various significance. Several recent reports consider such mutations, particularly concerning human diseases [41,42]. Silent mutations may affect translational efficiency and protein folding by changing codons read by tRNAs [43,44] and alter mRNA stability structure or splicing leading to variation in protein expression [45]. SNPs in the promoter regions could affect their activity and regulation producing changes in gene expression levels. Moreover, through different mechanisms, silent mutations may influence gene regulation, differences in mRNA and protein abundance, and proteins’ structure and functionality [45]. Untranslated regions (UTRs), which are localized at both ends of transcript (mRNA), make numerous conformational structures, tridimensional loops and hairpins that interact with numerous proteins and other functional and regulatory compounds like ribosomes or microRNA. UTRs are known to play crucial roles in the post-transcriptional regulation of gene expression. The main role of 5′UTR is controlling of translation efficiency as well as transcript stabilization while 3′UTR is mostly implicated in regulation of transcript stabilization, including modulation of the transport of mRNAs out of the nucleus [46,47]. Furthermore 3′UTR can be microRNA target sites, where microRNAs bind and regulate genetic expression. Therefore, SNPs or mutations in these regions might alter the existing target sites for microRNAs [48]. The importance of UTRs in regulating gene expression is underlined by the finding that mutations that alter the UTR can lead to serious pathology [49].

The role of TLR7 and TLR8 SNPs in SRLVs infection surely requires a comparison of infected and uninfected animals. Unfortunately, only 3 out of 32 tested goats were uninfected, so statistical analysis was not possible. However, it is worth pointing out that goats from tested flock were serologically examined several times over the past few years, and only these 3 goats remained always negative. Serological studies revealed that very young goats presented antibodies against SRLVs while these 3 goats, which have 3, 7 and 10 years old, were uninfected. This clearly indicates an implication of host factors that may restrict and control SRLVs replication. Because it was impossible to create control group with an equivalent number of animals to those of serologically positive and compare allele and genotype frequencies between infected and uninfected goats, we focused on investigating the association between TLR7 and TLR8 mutations and SRLVs proviral concentration, as important index of disease progression. It has been demonstrated that host control of infection with SRLVs, including provirus level, may have a genetic basis [10,50,51,52]. Genetic studies in humans have pointed to a role for TLR SNPs, especially TLR7, in HIV infection and their association with viral loads. The presence of the most frequent TLR7 polymorphisms, TLR7 Gln11Leu (rs179008), was associated with increased viral load and altered CD4 T cell counts during HIV infection [53,54] while polymorphisms in TLR7 (re179010) and TLR8 (rs3764880) may reduce the risk of disease by participating in inhibition of viral load leading to the slower progression of infection [28,55,56]. Our results revealed that 9 out 14 identified polymorphisms were significantly associated with the SRLVs proviral concentration. In particular, homozygous genotypes of three TLR7 SNPs (synonymous variant 1:50703293, 3′UTR variant 1:50701297 and 5′UTR variant 1:50718645) were observed in goats with lower provirus copy number as well as in seronegative animals. However, additional analysis is needed to confirm this association. In this analysis, the effort should be directed to goat’s samples collected at different time points to check the evolution of proviral load over the time.

The direct influence of the SNPs detected in this study on phenotypic features is unknown. Similar to what was showed for SRLVs infection in sheep [57], it can be assumed that during SRLVs infection in goats, TLR7 and TLR8 became activated, inducing production of antiviral cytokine, including IFN5-α, IL-6 and TNF-α production, which are critical in the development of antiviral immune response [21]. Activation of IFN type I pathway leads to expression of genes with IFN-sensitive response elements (ISREs) which fight viruses through different mechanisms. It has been confirmed that SRLVs contain ISRE and that IFN may modulate viral transcription promoting an antiviral state [14,58]. Therefore, we can speculate that SNPs detected in this study may affect the level of TLR7/8 expression, causing the differences in the production of downstream cytokines, like IFN-α, which finally can lead to the differences in SRLVs proviral concentration.

Studies on the association between TLR variants and HIV infection/disease progression suggested that they may be specific for different human populations [59]. Diversity reported between breed’s susceptibility/resistance to SRLVs [50,60] may suggest that SNPs detected in this study may be typical for goats of Carpathian breed. We especially focused on goats of this breed because the Carpathian goat is an ancient breed which in the 19th/20th century was present in Carpathian Mountain in Poland and then became an extinct breed. It emphasizes the need to select animals for higher resistance to infections. Considering the fact that animals belonging to this flock showed close to 100% seropositivity against SRLVs, the identification of genetic markers associated with SRLVs proviral load may be important for further breeding program. Removal of infected animals, especially those with high proviral load can limit the spread of the virus since such animals are highly efficient in shedding the virus [10].

In conclusion, this is the first report showing single nucleotide polymorphisms in genes encoding TLR7/8 in goats and the association between TLR7/8 polymorphisms and SRLVs proviral concentration in goats of Carpathian breed. Limited number of animals tested in this study and lack of possibility to compare infected and non-infected animals are undoubtedly limitations of this work. However, the obtained results suggest that SNPs of TLR7/TLR8 genes may induce differential innate immune response towards SRLVs affecting proviral concentration and thereby disease progression. These findings support a role of genetic variations in TLR7 and TLR8 in the course of infection with SRLVs and warrants further studies on TLR7/TLR8 polymorphisms.

## Figures and Tables

**Figure 1 animals-11-01908-f001:**
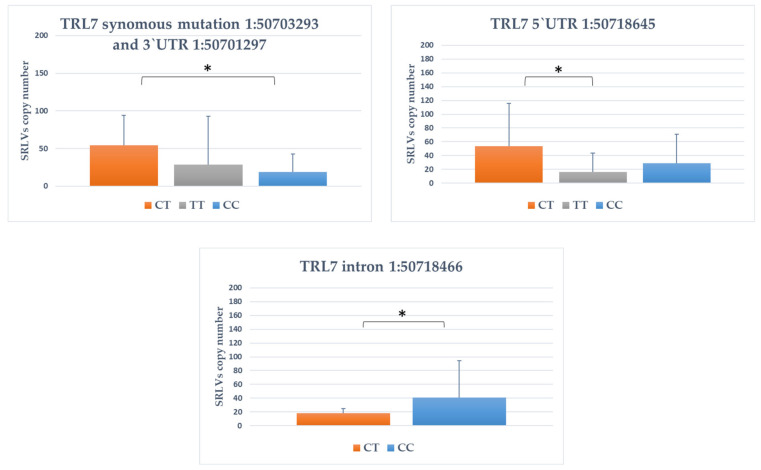
The association between genotype within TLR7 genes and SRLVs copy number. *: *p* < 0.05.

**Figure 2 animals-11-01908-f002:**
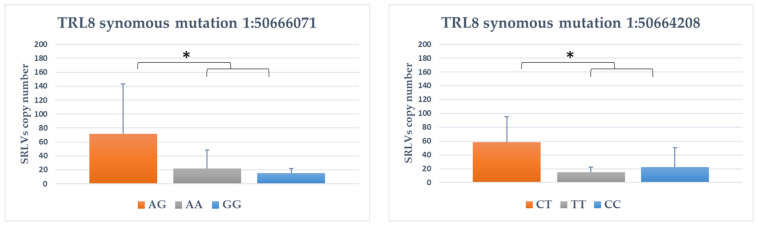
The association between genotype within TLR8 genes and SRLVs copy number. *: *p* < 0.05; **: *p* < 0.01.

**Table 1 animals-11-01908-t001:** Primers used for amplification of TLR7 and TLR8 polymorphisms.

Reference Sequence	Sequences of Primers (5′-3′)	Orientation	Size	Identified SNP Based on RNA-Seq Data
TLR7 ENSCHIG00000021012	GACATGAGGTTGCCCTGATTG	F	338 bp	LWLT01000021.1_50718645_C/T
	TATGATTTGGGTCCCTTCCCC	R		
	TCTCTGAGTTTCTTACCTTTGGGA	F	285 bp	LWLT01000021.1_50703293_T/C
	GCTGAGGTCCAGATGTCGC	R		
	CTTGATCGGCTCACCCATGC	F	297 bp	LWLT01000021.1_50702074_T/C
	AAGTGCAGTTCTTGGTGACTG	R		
	ACCTGCTTAAATATGGCTCTGG	F	418 bp	LWLT01000021.1_50701297_C/T
	TGGATGAGGCAACTTTTCTTTGG	R		
TLR8 ENSCHIG00000016625	GGGCATTTCTCAACCTCAAA	F	316 bp	LWLT01000021.1_50666071_A/G
	GTTAGCGAGCTTGGCAGACT	R		
	AATGCGCACTATTTCCGAAT	F	301 bp	LWLT01000021.1_50664682_A/G
	TCCTGGGCAAGTTAAGGAAG	R		
	GCAGCCTGATACACCTCGAT	F	424 bp	LWLT01000021.1_50664208_C/T LWLT01000021.1_50664064_C/T
	TGGGATGTGGACAGAGACCT	R		
	CTGATGTTGCTGGGAGTCCT	F	576 bp	LWLT01000021.1_50659559_C/T LWLT01000021.1_50659202_T/C
	TAGCTGAGAGGGGAATTGCC	R		

F—forward, R—reverse.

**Table 2 animals-11-01908-t002:** New polymorphisms within goat TLR7 and TLR8 genes identified using RNA-seq data.

	Location	Mutant Allele	Variant Type	Gene Reference	Transcript Reference	cDNA Position	Codons
TLR7							
LWLT01000021.1_50701297_C/T	LWLT01000021.1:50701297-50701297	T	3_prime_UTR_variant	ENSCHIG00000021012	ENSCHIT00000031299.1	4304	-
LWLT01000021.1_50702074_T/C	LWLT01000021.1:50702074-50702074	C	3_prime_UTR_variant	ENSCHIG00000021012	ENSCHIT00000031299.1	3527	-
LWLT01000021.1_50703293_T/C	LWLT01000021.1:50703293-50703293	C	synonymous_variant	ENSCHIG00000021012	ENSCHIT00000031309.1	2088	ggA/ggG
LWLT01000021.1_50718645_C/T	LWLT01000021.1:50718645-50718645	T	5_prime_UTR_variant	ENSCHIG00000021012	ENSCHIT00000031299.1	23	-
TLR8							
LWLT01000021.1_50659202_T/C	LWLT01000021.1:50659202-50659202	C	3_prime_UTR_variant	ENSCHIG00000016625	ENSCHIT00000024133.1	4708	-
LWLT01000021.1_50659559_C/T	LWLT01000021.1:50659559-50659559	T	3_prime_UTR_variant	ENSCHIG00000016625	ENSCHIT00000024133.1	4351	-
LWLT01000021.1_50664064_C/T	LWLT01000021.1:50664064-50664064	T	synonymous_variant	ENSCHIG00000016625	ENSCHIT00000024133.1	2460	gcG/gcA
LWLT01000021.1_50664208_G/A	LWLT01000021.1:50664208-50664208	A	synonymous_variant	ENSCHIG00000016625	ENSCHIT00000024133.1	2316	ttC/ttT
LWLT01000021.1_50664682_A/G	LWLT01000021.1:50664682-50664682	G	synonymous_variant	ENSCHIG00000016625	ENSCHIT00000024133.1	1842	ctT/ctC
LWLT01000021.1_50666071_A/G	LWLT01000021.1:50666071-50666071	G	synonymous_variant	ENSCHIG00000016625	ENSCHIT00000024133.1	453	aaT/aaC

**Table 3 animals-11-01908-t003:** New polymorphisms within goat TLR7 and TLR8 genes identified using Sanger sequencing method.

	Location	Mutant Allele	Variant Type	Gene Reference	Transcript Reference	cDNA Position
TLR7						
LWLT01000021.1_50718760_G/A	LWLT01000021.1:50718760-1: 50718760	G	Promoter	ENSCHIG00000021012	ENSCHIT00000031299.1	-
LWLT01000021.1_0718466_C/T	LWLT01000021.1:50717866-50717866	C	Intron 1	ENSCHIG00000021012	ENSCHIT00000031299.1	-
TLR8						
LWLT01000021.1_50659346_C/A	LWLT01000021.1: 50659346-50659346	C	3_prime_UTR_variant	ENSCHIG00000016625	ENSCHIT00000024133.1	4564
LWLT01000021.1_50659136_A/C	LWLT01000021.1: 50659136-50659136	A	3_prime_UTR_variant	ENSCHIG00000016625	ENSCHIT00000024133.1	4774

**Table 4 animals-11-01908-t004:** The genotype distribution for SNPs detected in TLR7 gene.

SNP	Genotypes			Alleles		HWE *p* Value
	CC	CT	TT	C	T	
SV T/C 1:50703293	11 (31%)	16 (46%)	5 (23%)	38 (0.54)	26 (0.46)	ns
	CC	CT	TT	C	T	
3′UTR C/T 1:50701297	5 (23%)	16 (46%)	11 (31%)	38 (0.54)	26 (0.46)	ns
	CC	CT	TT	C	T	
5′UTR C/T 1:50718645	5 (16%)	17 (53%)	10 (31%)	27 (0.42)	37 (0.58)	ns
	CC	CT	TT	C	T	
INTRON1 C/T 1:50718466	29 (91%)	2 (6%)	1 (3%)	58 (0.94)	6 (0.06)	0.00001

HWE—Hardy-Weinberg equilibrium; ns—not significant.

**Table 5 animals-11-01908-t005:** The haplotypes frequency of TLR7 and TLR8 genes.

	Haplotype	Frequency	S.E.
TLR7			
1	T/T/C/T/A/C	51%	0.003
2	C/T/T/C/A/C	29%	0.004
3	C/T/T/C/A/T	10%	0.004
4	T/T/T/C/A/C	4%	0.002
5	T/T/C/C/A/C	4%	0.0003
6	T/T/C/T/G/C	1.20%	0.000
7	T/T/T/C/A/T	<1%	0.002
8	T/T/C/T/A/T	<1%	0.003
TLR8			
1	C/C/C/G/A/A/C/A	29.70%	0.010
2	C/T/T/A/A/G/A/A	19.90%	0.003
3	C/C/C/G/A/A/C/C	14.80%	0.008
4	C/T/C/G/A/A/C/A	13.40%	0.007
5	C/T/C/G/A/A/C/C	13.30%	0.008
6	C/T/T/A/A/G/C/A	5.70%	0.001
7	C/T/C/G/G/A/C/A	2.70%	0.003
8	C/C/T/A/A/G/A/A	<1%	0.002
9	C/T/C/G/G/A/C/C	<1%	0.003
10	C/T/T/A/A/G/A/C	<1%	0.002

SNPs order: TLR7-LWLT01000021.1-50701297; 50702074; 50703293; 50718645; 50718760; 50718466; TLR8–LWLT01000021.1 -50659202; 50659559; 50664064; 50664208; 50664682; 50666071; 50659346; 50659136.

**Table 6 animals-11-01908-t006:** The genotype distribution for SNPs detected in TLR8 gene.

SNP	Genotypes			Alleles		HWE *p* Value
	AA	GA	GG	A	G	
SV A/G 1:50666071	18 (56%)	11 (34%)	3 (10%)	47 (0.73)	17 (0.27)	ns
	CC	CT	TT	G	A	
SV G/A 1:506642208	18 (56%)	11 (34%)	3 (10%)	47 (0.73)	17 (0.27)	ns
	CC	CT	TT	C	T	
SV C/T 1:50664064	18 (56%)	11 (34%)	3 (10%)	47 (0.73)	17 (0.27)	ns
	AA	AC	CC	A	C	
3′UTR A/C 1:50659136	17 (55%)	10 (32%)	4 (13%)	42 (0.71)	18 (0.29)	ns
	CC	AC	AA	C	A	
3′UTR C/A 1:50659346	17 (55%)	13 (42%)	1 (3%)	47 (0.76)	15 (0.24)	ns
	CC	TC	TT	C	T	
3″UTR C/T 1:50659559	15 (48%)	13 (42%)	3 (10%)	43 (0.69)	19 (0.31)	ns

HWE—Hardy-Weinberg equilibrium; ns—not significant.

## Data Availability

RNA-seq data are available under GEO GSE168160 accession number. Rest of data generated during and/or analysed during the current study are available from the corresponding author on reasonable request.

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
