# Peer review of "Single Nucleotide Polymorphisms in Genes Encoding Toll-Like Receptors 7 and 8 and Their Association with Proviral Load of SRLVs in Goats of Polish Carpathian Breed"

_animals, 2021, doi:10.3390/ani11071908_

Round 1
Reviewer 1 Report
The authors have addressed my concerns.
Reviewer 2 Report
The manuscript “Single nucleotide polymorphisms in genes encoding toll-like receptors 7 and 8 and their association with proviral load of SRLVs in goats of Polish Carpathian breed”, Animals ID 1183793, is now acceptable for publication in this revised version.
I appreciate the notes of the authors and the revisions done contributed to make the text clearer; in particular statistical analysis related to proviral load and considerations and explanations reported in the discussion section.
This manuscript is a resubmission of an earlier submission. The following is a list of the peer review reports and author responses from that submission.
Round 1
Reviewer 1 Report
This work identified variants in TLR7 and TLR8 and tested for association with SRLV load. Some variants were significantly associated with load, which confirms in goats what has been observed in sheep. The sample size was quite small, but the combination of biological involvement for these genes in other lentivirus infections and the significant association suggest this work adds value to the literature. However, I would like to understand some factors better to be sure this is true.
Critical revisions:
- Account for goat age? Many prior studies identified animal age as an important covariate for SRLV proviral load. The authors could use a simple model with age as a covariate to account for age or age groups to better assess association. Or if age is not a significant factor for load in this animal set, please state your preliminary test and its results to justify not including age as a covariate.
- The authors mention microRNA binding as a mutation mechanism. Are either of the 3’ UTR variants predicted to affect microRNA binding? In silico testing alleles for creation or destruction of microRNA sites and describing results would strengthen the study.
Minor revisions:
- Haplotype analyses (e.g. using FastPHASE) would be helpful for describing the multiple variants. I suspect they are in strong linkage disequilibrium and form only a small number of haplotypes in this dataset.
- In line 127, “spam” should be corrected to “span”.
Author Response
We would like thank the reviewer for his comments on our manuscript. We have acted upon the suggestions provided by the reviewer and alterations were included in the updated version of the manuscript.
This work identified variants in TLR7 and TLR8 and tested for association with SRLV load. Some variants were significantly associated with load, which confirms in goats what has been observed in sheep. The sample size was quite small, but the combination of biological involvement for these genes in other lentivirus infections and the significant association suggest this work adds value to the literature. However, I would like to understand some factors better to be sure this is true.
Critical revisions:
- Account for goat age? Many prior studies identified animal age as an important covariate for SRLV proviral load. The authors could use a simple model with age as a covariate to account for age or age groups to better assess association. Or if age is not a significant factor for load in this animal set, please state your preliminary test and its results to justify not including age as a covariate.
Re: The correlation between proviral load and age of analyzed goats has been estimated and was not statistically significant. Thus age was not included in the model, but clearly pinpointed that such preliminary analysis was done and this factor did not affect the results.
- The authors mention microRNA binding as a mutation mechanism. Are either of the 3’ UTR variants predicted to affect microRNA binding? In silico testing alleles for creation or destruction of microRNA sites and describing results would strengthen the study.
Re: According reviewer’s comments, we try to perform in silico analysis of miRNA and both targeted genes taking into account different alleles. We used TarBase v.8 and sRNAtoolbox to predict potential effect of 3’UTR variant. Due to the poor annotation of goats miRNA we used human and mouse miRNA structure which target TLRs genes. Nevertheless, the small number of identified miRNAs associated with TLR7 and TLR8 genes, and no available information for goats result that we were not able to carry out such analysis.
Minor revisions:
- Haplotype analyses (e.g. using FastPHASE) would be helpful for describing the multiple variants. I suspect they are in strong linkage disequilibrium and form only a small number of haplotypes in this dataset.
Re: According to your suggestion, we performed the haplotype analysis using Phase software. Moreover, Haploview program was used to generate haplotype block structures and calculate LD values between SNPs.
- In line 127, “spam” should be corrected to “span”.
Re: It has been corrected.
Reviewer 2 Report
The manuscript “Single nucleotide polymorphisms in genes encoding toll-like receptors 7 and 8 and their association with proviral load of SRLVs in goats of Polish Carpathian breed” is of interest because data on TLR 7 and 8 in goats and their association with SRLV infection are still limited; nevertheless in my opinion this paper has relevant flaws listed below:
- the number of goats analyzed is very small; only 32 goats were analyzed, and this is not enough to perform an association study.
- authors have only a few negative samples so decided to associate polymorphisms of TLR 7 and 8 with proviral load, but how did they classify proviral loads? How did they define high proviral loads?
- genetic nomenclature of polymorphisms in regulatory regions is not correct: at this purpose the Authors should follow guidelines for variation nomenclature (https://www.hgvs.org/content/guidelines)
- with such a small number of goats analyzed and the absence of negative samples, considerations reported in the discussion should be considered as mere speculations.
Moreover, obtained data related to goats of Polish Carpathian breed (local breed with a limited number of animals, covered by the genetic resources conservation program) could be of interest only for this breed especially if a genetic bottleneck event happened in the past.
For all the reason listed above the manuscript can not be accepted in its present form.
Author Response
We would like thank the reviewer for his comments on our manuscript.
The manuscript “Single nucleotide polymorphisms in genes encoding toll-like receptors 7 and 8 and their association with proviral load of SRLVs in goats of Polish Carpathian breed” is of interest because data on TLR 7 and 8 in goats and their association with SRLV infection are still limited; nevertheless in my opinion this paper has relevant flaws listed below:
- the number of goats analyzed is very small; only 32 goats were analyzed, and this is not enough to perform an association study.
Re: In our study we focused on identification of SNPs in genes encoding TLR7 and TLR8 in goats representing only Carpathian breed. We focused on goats of Carpathian breed because it represents a remnant population of an ancient breed and thus analysis could be important for its conservation. This information was added in the introduction. Moreover, it is first results presented concerning association between TLRs genes and proviral load in goat as a species. This breed is represented only by one flock of 32 goats, which were tested in this study so it is impossible to perform on additional study including more animals. Moreover, it was extremally important to us to study the animals which represent the same breed and which were maintained at the same environmental and feeding conditions.
authors have only a few negative samples so decided to associate polymorphisms of TLR 7 and 8 with proviral load, but how did they classify proviral loads? How did they define high proviral loads?
Re: To perform the association analysis between genotypes and proviral load, we used T-test where SNP was a classification variable and virus copies values were as analysis variable. In such linear variable as proviral load, the means presented specific significant differences between animals within given genotype groups. So proviral loads values results from statistical grouping of animals. The definitio „high and low” proviral loads reflects statistical means.
- genetic nomenclature of polymorphisms in regulatory regions is not correct: at this purpose the Authors should follow guidelines for variation nomenclature (https://www.hgvs.org/content/guidelines)
Re: We agree that the showed nomenclature is long and thus hard to follow, but such nomenclature (with scaffold ID and nucleotide location on the chromosome) implemented from reference genome used and variant calling approach. They were not created by us, but results from valid nomenclature in reference genome. We submit the newly identified SNPs to EVA database and the will revived rs#. Thus, we think that introducing even more nomenclature (third) would create confusion and generate additional errors. Of course we can modify it but we hypothesized that it will be confusing to the reader.
- with such a small number of goats analyzed and the absence of negative samples, considerations reported in the discussion should be considered as mere speculations. Moreover, obtained data related to goats of Polish Carpathian breed (local breed with a limited number of animals, covered by the genetic resources conservation program) could be of interest only for this breed especially if a genetic bottleneck event happened in the past.
Re: Limited number of animals used in this study is undoubtedly a limitation of this work and this information was included at the end of discussion. This study is a preliminary research which gave promising results and in the future we will extended such analysis on other goats breeds as well as sheep. As we mentioned above, we focused only on goats of Carpathian breed which is extinct breed in Poland. SNPs detected in this study may be typical for goats of Carpathian breed and this information is included in discussion (L 408). Serological tests of goats from analyzed flock were performed several times and only 3 animals were always negative that’s why it was impossible to create control group with an equivalent number of animals to those of serologically positive and compare allele and genotype frequencies between infected and uninfected goats. Additionally, in our report, we used statistical approach which does not require the negative controls - identified SNPs were a classification variable and virus copies values were as analysis variable. In such linear variable as proviral load, the means presented specific significant differences between animals within given genotype groups.